# On-Site Dual Detection of Airborne *Acinetobacter baumannii* and Its Carbapenem-Resistant Gene *bla*_OXA-23_ Using a One-Pot Visual LAMP-CRISPR/Cas12a-Based Platform

**DOI:** 10.3390/microorganisms13050976

**Published:** 2025-04-24

**Authors:** Huijun Lu, Tong Zhang, Wei Huang, Jinhui Zhu, Haoran Qin, Xi Chen, Wang Zhao, Guodong Sui

**Affiliations:** 1Shanghai Key Laboratory of Atmospheric Particle Pollution and Prevention (LAP3), Department of Environmental Science & Engineering, Fudan University, Shanghai 200433, China; 2Department of Clinical Laboratory, Shanghai East Hospital, School of Medicine, Tong Ji University, Shanghai 200120, China; 3Shanghai Institute of Infectious Disease and Biosecurity, Shanghai 200433, China; 4Department of Medical Microbiology and Parasitology, School of Basic Medical Sciences, Fudan University, Shanghai 200032, China; 5Jiangsu Collaborative Innovation Center of Atmospheric Environment and Equipment Technology (CICAEET), Nanjing University of Information Science & Technology, Nanjing 210044, China

**Keywords:** the CLC platform, airborne pathogen, carbapenem-resistant gene, nucleic acid detection

## Abstract

*Acinetobacter baumannii (A. baumannii)*, a very common pathogen, poses a significant public health threat due to its antibiotic resistance and long survival in healthcare environments. Both *A. baumannii* and carbapenem-resistant *A. baumannii* (CRAB) can spread through the air, increasing infection risks. Therefore, monitoring their presence in the air is of great significance, especially in hospitals. Herein, we developed a Chelex-100-LAMP-CRISPR/Cas12a (CLC) platform including DNA release and nucleic acid test. Combined with a wet cyclone sampler, the platform can detect airborne *A. baumannii* and its most common carbapenem-resistant gene, *bla*_OXA-23_, within 70 min. This CLC platform has also been proven to have a detection limit of 6 × 10^2^ CFU of CRAB per test through simulated air samples. Moreover, this platform was also used to test five actual air samples from a tertiary hospital, and the results achieved perfect concordance with sequencing data, validating the platform’s accuracy and reliability. Therefore, the CLC platform showed great potential for the rapid, on-site detection of airborne *A. baumannii* and its carbapenem-resistant gene *bla*_OXA-23_, offering a valuable tool for infection control in healthcare environments.

## 1. Introduction

*Acinetobacter baumannii* has emerged as a critical global health threat due to its antibiotic resistance and ability to cause severe nosocomial infections [1]. As an opportunistic pathogen, *A. baumannii* predominantly infects critically ill patients [2], leading to a wide range of nosocomial infections. These include respiratory infections such as ventilator-associated pneumonia, bloodstream infections like central-line-associated bacteremia, and localized infections of the skin, soft tissues, surgical sites, and the urinary tract [3,4,5]. Compounding the problem, carbapenem-resistant *A. baumannii* (CRAB) has become increasingly prevalent and is now classified as a critical priority pathogen by the WHO [6]. Carbapenem resistance in *A. baumannii* is mediated by multiple mechanisms, including enzymatic degradation by carbapenemases, efflux pump overexpression, and porin modifications [7]. Among these mechanisms, the production of carbapenemases, particularly OXA-type enzymes, represents the most prevalent mechanism. Notably, OXA-23, a class D carbapenemase encoded by *bla*_OXA-23_ gene, is one of the most widely distributed carbapenemases in CRAB [8].

Several studies, particularly those conducted in hospital environments, have provided evidence supporting the airborne transmission of *A. baumannii* [2]. Within a ward, *A. baumannii* can spread from infected patients to the surrounding environment and other susceptible patients via aerosol droplets. Wong et al. [9] used whole-genome sequencing to investigate the clonality and the genetic relatedness of *A. baumannii* isolates from patients, the environment, and air samples. Their results showed that the *A. baumannii* strains from these sources clustered together in the maximum likelihood tree, providing strong evidence for airborne dispersal. Furthermore, *A. baumannii* can survive on dry surfaces for a relatively long period, ranging from 21 to 33 days. This suggests that it has strong environmental tolerance and may remain viable in the airborne state for a prolonged period, thereby increasing the risk of transmission [10,11]. Given the serious implications of *A. baumannii* and *CRAB* infections, as well as the potential risk of airborne transmission in hospital environments, rapid detection of these pathogens in air samples is critically important for infection control.

Air sample collection represents the first and one of the most crucial steps in detecting airborne microbes, with various methods developed to address different environmental and sampling conditions. These methods include gravity [12], electrostatic attraction [13], thermal precipitation [14], impaction [15], cyclone [16], impingement [15], and filtration [15]. Among these, wet cyclone sampling is commonly employed for rapid detection of airborne pathogens and real-time monitoring of microbe aerosols. For example, Puthussery et al. integrated a high-flow wet cyclone air sampler into a micro-immunoelectrode biosensor for real-time monitoring of airborne SARS-CoV-2 [17], while Lee et al. combined a wet cyclone air sampler with a DC impendence microfluidic cytometer for bioaerosol monitoring [18]. Moreover, wet cyclone samplers have demonstrated excellent performance in trapping microbes, followed by nucleic acid detection methods like PCR. Watt et al. employed wet cyclone air sampling and qPCR technology to detect naturally aerosolized *Actinobacillus pleuropneumoniae* in pig farms [19].

Various methods have been developed for the detection of airborne bacteria. Traditional culture-based methods, widely used in environmental surveillance due to their simplicity and low cost, are limited by their time-consuming and labor-intensive nature. Furthermore, only a small fraction (approximately 10%) of environmental microbes are culturable, significantly restricting their applicability [12,20]. Currently, PCR and real-time quantitative PCR (qPCR) technologies are commonly employed for microbial detection [21,22]. However, these methods rely on expensive and precise thermal cycling instruments, which hinder their use in on-site applications. As an alternative, loop-mediated isothermal amplification (LAMP) has emerged as a promising alternative for detecting microbes [23,24,25]. Although LAMP has been relatively less applied in the detection of airborne microbes [26,27], it offers rapid, cost-effective, and easy-to-use detection without the need for complex thermal cycling equipment, making it particularly suitable for field applications. Moreover, with the development of clustered regularly interspaced short palindromic repeats (CRISPR) technology, recent studies have demonstrated that coupling LAMP with CRISPR-assisted detection enhances sensitivity and specificity, offering significant improvements in isothermal detection [28,29]. Therefore, it is worth exploring the potential of LAMP-CRISPR in the detection of airborne microbes.

In this study, we developed a novel detection platform. This platform integrates a Chelex-100-based lysis method with one-pot LAMP-CRISPR/Cas12a technology, named the CLC platform, achieving visible dual identification of *A. baumannii* and *bla*_OXA-23_. When combined with a wet cyclone sampler, the CLC platform allows for the detection of airborne *A. baumannii* and *bla*_OXA-23_ within 70 min, including sampling (5 min), DNA release (5 min), and nucleic acid testing (55 + 5 min). To the best of our knowledge, this represents the first application of the one-pot LAMP-CRISPR/Cas12a technology for the detection of airborne *A. baumannii* and its carbapenem-resistant gene *bla*_OXA-23_, providing a novel and effective approach for the environmental surveillance of pathogenic microorganisms.

## 2. Materials and Methods

### 2.1. Materials and Reagents

The *A. baumannii* (ATCC 19606) control strain was purchased from the American Type Culture Collection (ATCC). The clinical and environmental isolates of *A. baumannii* were kindly provided by Shanghai East Hospital and Zhongshan Hospital. Other bacterial strains, including *Staphylococcus aureus* (ATCC 25923), *Staphylococcus epidermidis* (ATCC 12228), *Streptococcus pneumoniae* (ATCC 49619), *Escherichia coli* (ATCC 35218), *Pseudomonas aeruginosa* (ATCC 27853), and *Klebsiella pneumoniae* (ATCC 13883), were obtained from ATCC. The fungal strain *Candida albicans* (ATCC 10231) was also obtained from ATCC, while *Aspergillus fumigatus* (CICC 41022) was acquired from the China Center of Industrial Culture Collection (CICC). *Cryptococcus neoformans* was a clinical isolate that was kindly provided by Zhongshan Hospital and identified by MALDI-TOF mass spectrometry. Bst DNA polymerase (large fragment), dNTP_S_, the T7 High Yield RNA Transcription Kit, and Recombinant RNase Inhibitor (RRI) were purchased from Novoprotein Biotechnology Co., Ltd. (Suzhou, China). SYBR Green I was acquired from Thermo Fisher Scientific (Waltham, MA, USA). Lba Cas12a (Cpf1) was purchased from New England Biolabs (Ipswich, UK). Primers were synthesized in General biol (Chuzhou, China). Mineral oil was obtained from Amresco (Ohio, USA). The TIANamp Bacteria DNA kit was purchased from Tiangen Biotechnology Co., Ltd. (Beijing, China). Nutrient agar was obtained from Sangon Biotech (Shanghai, China).

### 2.2. Design and Selection of Primers and crRNAs

The sequence of intergenic spacer (ITS) region, located between the 16S and 23S rRNA genes of *A. baumannii* (accession number: AY601823), was retrieved from the National Center for Biotechnology Information (NCBI) database (http://www.ncbi.nlm.nih.gov (accessed on 12 December 2023)). Similarly, the sequence of the *bla*_OXA-23_ gene, which encodes the OXA-23 carbapenemase enzyme, was obtained from the Comprehensive Antibiotic Resistance Database (CARD) (https://card.mcmaster.ca (accessed on 12 December 2023)). Both sequences were used to design LAMP primers by PrimerExplorer V5 (https://primerexplorer.eiken.co.jp/lampv5e/index.html (accessed on 15 December 2023)). Furthermore, crRNAs were designed based on the sequences of the target genes amplified by LAMP. The UNAFold website (http://www.unafold.org/index-top.php (accessed on 20 January 2024)) was used to predict the secondary structures of these crRNAs. CrRNAs with optimal structural characteristics were selected and synthesized for further research. The selection criteria are detailed in the Appendix A).

### 2.3. Establishment and Optimization of One-Pot LAMP-CRISPR/Cas12a Assay

The one-pot LAMP-CRISPR/Cas12a assay consisted of a 25 μL LAMP reaction mixture, a 25 μL CRISPR/Cas12a reaction mixture, and 20 μL of mineral oil. The LAMP reaction mixture contained 1× Bst LF Buffer with Mg^2+^, 1 mM dNTPs, 6 mM Mg^2+^, 800 mM betaine, 8 U Bst DNA polymerase Large Fragment, 1.6 μM each of inner primers FIP and BIP, 0.2 μM each of outer primers F3 and B3, 0.8 μM each of loop primers LF and LB, double distilled water (ddH_2_O), and 5 μL of template. These components were added to the bottom of the reaction tube, followed by gentle overlay with 20 μL of mineral oil, the optimal volume of which was determined through preliminary optimization (Appendix A). The CRISPR/Cas12a reaction mixture contained RNase-free water, 2× NEB buffer r2.1, 50 U RRI, 50 nM to 200 nM crRNA, 60 nM to 300 nM Lba Cas12a, 200 nM to 2 μM FAM- or ROX-labeled reporter probes. This CRISPR/Cas12a reaction mixture was placed on the inner surface of the lid of the reaction tubes. The tubes were first incubated in a heat block at 63 °C for 55 min to complete LAMP reaction. Subsequently, the CRISPR/Cas12a reaction mixture was mixed with the LAMP reaction product by brief shaking and centrifugation. After the CRISPR/Cas12a-mediated cleavage reaction at 37 °C, the fluorescence signal was detected by ABI 7500 real-time PCR systems or visualized under UV light (312 nm). The one-pot LAMP-CRISPR/Cas12a assay was optimized by determining the optimal concentration of Lba Cas12a, crRNAs, and reporter probes, as detailed in the Appendix A.

### 2.4. Specificity Test of the One-Pot LAMP-CRISPR/Cas12a Assay

To evaluate the specificity of the one-pot LAMP-CRISPR/Cas12a assay for detecting *A. baumannii*, the control strain ATCC 19606, eight clinical isolates of CRAB, two environmental isolates of CRAB, and nine other common pathogenic bacteria were used as templates.

To evaluate the specificity of the one-pot LAMP-CRISPR/Cas12a assay targeting the *bla*_OXA-23_ gene, a panel of templates was tested, including eight clinical isolates previously confirmed by sequencing to harbor the *bla*_OXA-23_ gene, ATCC19606, and three carbapenem-susceptible *A. baumannii* (CSAB) strains lacking the *bla*_OXA-23_ gene. Each template was tested at a concentration of approximately 1 μg, and all experiments were conducted in triplicate to ensure reproducibility and reliability.

### 2.5. Sensitivity Test of the One-Pot LAMP-CRISPR/Cas12a Assay

To determine the sensitivity of the proposed assay, genomic DNA was extracted from ATCC 19606 and a clinical isolate harboring the *bla*_OXA-23_ gene using the TIANamp Bacteria DNA kit. The concentration of the extracted DNA was quantified by Qubit-Flex fluorometer with a dsDNA (broad range) kit. Then, the DNA from ATCC 19606 and the clinical isolate carrying the *bla*_OXA-23_ gene were diluted in 10-fold intervals from ng/μL to fg/μL for the sensitivity evaluation of the *A. baumannii* and the *bla*_OXA-23_ assays, respectively. ddH_2_O was used as the negative control template. All tests were performed in triplicate.

### 2.6. Development of a Rapid DNA Release Method for A. baumannii

To enable rapid and cost-effective detection of *A. baumannii* and its antibiotic resistance gene *bla*_OXA-23_, three DNA extraction methods were evaluated: (a) Thermolysis: bacteria were dissolved in 100 μL phosphate-buffered saline (PBS) and then subjected to thermolysis at 63 °C; (b) Chelex-100: bacteria were dissolved in 100 μL of 5% Chelex-100 and then subjected to thermolysis at 63 °C; and (c) Chelex-100-Triton X-100: bacteria were dissolved in 100 μL of a mixture of 5% Chelex-100 and 1% Triton X-100 in TE buffer, followed by thermolysis at 63 °C. To assess the efficiency of nucleic acid release, low bacterial load (1000 CFU) and moderate bacterial load (100,000 CFU) of *A. baumannii* were separately treated using these three methods, with each test performed in triplicate. After treatment, 5 μL of the supernatant was directly added to the LAMP reaction mixture (prepared as described in Section 2.3), containing 5× SYBR Green I. After a 55 min amplification, melt curve analysis was performed to confirm positive results.

### 2.7. Evaluation of the CLC Platform Using Simulated Air Samples

The rapid DNA release method based on Chelex-100 was integrated with the one-pot LAMP-CRISPR/Cas12a assay into a platform, designated as the CLC platform, for the detection of *A. baumannii* and the *bla*_OXA-23_ gene. When coupled with a wet cyclone sampler, the CLC platform can be used to detect *A. baumannii* and the *bla*_OXA-23_ gene in air samples. To validate the capability of the CLC platform in testing air samples, aerosol simulation experiments were conducted in a 1 m^3^ bioaerosol test chamber (ZR-1042 type, Junray, Qingdao Zhongrui intelligent instrument, Co., Ltd., Qingdao, China). Prior to the experiments, all items within the chamber and the chamber itself were thoroughly sterilized using a 0.5% sodium hypochlorite solution, which was sprayed onto all surfaces and incubated for 30 min at room temperature, followed by thorough wiping with sterile distilled water to remove any residual disinfectant. The simulation experiments were performed as follows: (a) An aerosol of CRAB carrying the *bla*_OXA-23_ gene was generated within the chamber using a bacterial suspension. Five air fans operating at approximately 500 rpm were utilized to ensure uniform distribution of the aerosol. The aerosol was generated at a rate of 10 L/min for 5 min. (b) Following aerosol generation, airborne CRAB was immediately collected into 4 mL of 1X PBS using a wet cyclone sampler (Changhe Biotechnology Co., Ltd., Suzhou, China) at a flow rate of 300 L/min for 5 min. (c) After aerosol collection, 750 μL of each collected sample was plated onto nutrient agar plates for bacterial quantification, and the remaining portion of the sample was analyzed using the proposed CLC platform.

### 2.8. Feasibility Evaluation of the CLC Platform for Actual Air Sample Detection

The same wet cyclone sampler was used to collect air samples from a tertiary hospital using the same sampling methods as described in Section 2.7. Following collection, half of each sample was sent to KingMed Diagnostics company (Guangzhou, China) for target next-generation sequencing (tNGS) using the Respiratory Pathogen Detection Kit (KS608-100HXD96, KingCreate, Guangzhou, China), while the remaining portion of each sample was analyzed by Sanger sequencing and the proposed CLC platform in parallel. Detailed protocols for Sanger sequencing analysis are provided in Appendix A.

Other methods are listed in the Appendix A.

## 3. Results

### 3.1. Design and Workflow of the CLC Platform for Rapid Detection of Airborne A. baumannii and bla_OXA-23_ Gene

The workflow of the CLC platform is depicted in Figure 1. Aerosols were collected using a wet cyclone sampler, followed by bacterial lysis with 5% Chelex-100 at 63 °C for 5 min. Subsequently, one-pot LAMP-CRISPR/Cas12a assays were utilized to detect *A. baumannii* and carbapenem resistance gene *bla*_OXA-23_. For these assays, a FAM-labeled reporter probe paired with ITS-crRNA and a ROX-labeled reporter probe paired with *bla*_OXA-23_-crRNA were introduced to specifically identify *A. baumannii* and the *bla*_OXA-23_ gene, respectively. The sequences of LAMP primers and crRNAs used in this study are provided in Appendix A. The detection of *A. baumannii* was indicated by green fluorescence, while the presence of the *bla*_OXA-23_ gene was indicated by purplish-red fluorescence. Consequently, the simultaneous observation of green fluorescence from the *A. baumannii* assay and purplish-red fluorescence from the *bla*_OXA-23_ assay strongly indicated the presence of CRAB in the tested sample.

The entire detection process of the CLC platform for airborne pathogen detection can be completed within 70 min, including 5 min for sampling, 5 min for chelex-100-based rapid DNA release, and 55 min + 5 min for one-pot LAMP-CRISPR/Cas12a assay (Figure 1). This platform utilizes simple and portable devices, such as a wet cyclone sampler, heat block and UV lamp, enabling fast on-site detection.

### 3.2. Optimization of One-Pot LAMP-CRISPR/Cas12a Assay

We optimized the one-pot LAMP-CRISPR/Cas12a assay in terms of reaction time, Cas12a and crRNA concentration, and reporter probe concentration to achieve accurate and reliable detection. About 1 ng of *A. baumannii* genomic DNA was used as the template for LAMP, followed by CRISPR/Cas12a-mediated cleavage reaction under various conditions. The optimization results for reaction time and Cas12a concentration experiments are presented in Figure 2a. Higher concentrations of Cas12a enzyme and crRNA yielded stronger fluorescence signals, and the fluorescence intensity also increased progressively over time. However, after 5 min of CRISPR/Cas12a-mediated cleavage, the fluorescence signal intensity in the reaction system containing 50 nM Cas12a enzyme and 50 nM crRNA was high enough for visual detection under UV light (312 nm). From both economic and practical perspectives, lower concentrations of Cas12a enzyme and shorter reaction time were considered more suitable for on-site testing. Therefore, a concentration of 50 nM Cas12a enzyme and a cleavage time of 5 min were selected for subsequent detection experiments.

We further optimized the concentration of crRNA in the one-pot LAMP-CRISPR/Cas12a assay. The fluorescence signals in reactions containing 100 nM crRNA were significantly higher than those with 25 nM crRNA (*p* < 0.05). However, one-way ANOVA statistical analysis revealed no significant difference among the groups using 25 nM, 50 nM, and 75 nM crRNA (*p* > 0.05) (Figure 2b). Notably, even at 25 nM crRNA, the fluorescence signals were sufficiently vivid to be visually detected under UV light (312 nm). Considering the goal of developing a visual detection method, 25 nM crRNA was selected for subsequent experiments.

The concentration of the reporter probe was critical for the performance of the assay, as the fluorescence signal was generated through the cleavage of the probe by the cis-cleavage activity of Cas12a. To optimize the assay conditions, various concentrations of the FAM-labeled reporter probe were tested. The fluorescence signal intensity was significantly higher in reactions containing 1 μM FAM-labeled reporter probe compared to other concentrations (Figure 2c). Therefore, 1 μM FAM-labeled reporter probe was used in subsequent experiments.

To better visually distinguish between the fluorescence signals of the *A. baumannii* and *bla*_OXA-23_ assays, the concentration of the ROX-labeled reporter probe, replacing the FAM-labeled reporter probe, was further optimized in the *bla*_OXA-23_ assay. Optimization experiments revealed that a 1 μM concentration of the ROX-labeled reporter probe produced the highest fluorescence signal (Appendix A). Therefore, the final concentration in the optimized one-pot LAMP-CRISPR/Cas12a assay were established as 50 nM for Cas12a, 25 nM for crRNA, and 1 μM for the reporter probe. These conditions were selected to ensure optimal assay performance and to enable clear visual discrimination between the two targets.

### 3.3. Specificity and Feasibility of the One-Pot LAMP-CRISPR/Cas12a Assay

The specificity and feasibility of the one-pot LAMP-CRISPR/Cas12a assay were evaluated using the standard strain of *A. baumannii* (ATCC 19606), eight clinical isolates of CRAB, two environmental isolates of CRAB (sourced from ventilator interfaces), and nine common hospital-acquired infection pathogens. As shown in Figure 3a, all samples containing *A. baumannii* yielded high green fluorescence signals, indicating positive results. In contrast, other samples showed almost no fluorescence signals, which were indistinguishable by the naked eye. These results confirmed the feasibility and high specificity of the proposed assay for *A. baumannii* detection. In Figure 3b, the eight clinical CRAB isolates harboring the *bla*_OXA-23_ gene showed strong purplish-red fluorescence signals, with no false positives or cross-reactivity observed in non-target genes from CSAB samples or the ATCC 19606 strain. These results demonstrate that the *bla*_OXA-23_ assay is highly reliable for the specific detection of the *bla*_OXA-23_ gene.

### 3.4. Sensitivity of the One-Pot LAMP-CRISPR/Cas12a Assay

Sensitivity is also an important parameter for evaluating assay performance. To determine the sensitivity of the established assays, genomic DNA extracted from *A. baumannii* (ATCC 19606) and a clinical isolate harboring the *bla*_OXA-23_ gene were used. For the *A. baumannii* assay, significantly increased fluorescence values were observed from 3.03 ng/μL to 303 fg/μL genomic DNA, but not from 30.3 fg/μL to 3.03 fg/μL genomic DNA, or for NC (Figure 4a,b). Similarly, for the *bla*_OXA-23_ assay, significantly elevated fluorescence signals were observed from 1.74 ng/μL to 174 fg/μL genomic DNA, but not from 17.4 fg/μL to 1.74 fg/μL genomic DNA, or for NC (Figure 4d,e). Consistent results were observed by the naked eye under UV light (312 nm) (Figure 4c,f). The limit of detection (LOD) was calculated using the following equation: DNA copy number (copies/μL) = [6.02 × 10^23^ × DNA concentration (ng/μL) × 10^−9^ × *n*] / [genomic DNA length (bp) × 660], where *n* represents the copy number of the target gene. Based on this equation, the LOD for the *A. baumannii* assay using extracted DNA was equivalently 415 copies/μL, and the LOD for the *bla*_OXA-23_ assay was 79 copies/μL, respectively. These results demonstrate that the developed assay exhibits high sensitivity and can be performed without specialized equipment, significantly enhancing its feasibility for on-site applications in resource-limited settings.

### 3.5. Evaluation of Rapid DNA Release Methods for A. baumannii

Chelex-100 is widely used in nucleic acid extraction due to its ability to bind divalent metal ions, such as magnesium (Mg^2+^) and calcium (Ca^2+^), which are essential cofactors for nucleases [30]. Combined with the heating procedure, Chelex-100 not only prevented nucleic acid degradation by nucleases but also facilitated nucleic acid release [31,32]. In this study, three methods were evaluated for the rapid lysis of *A. baumannii*, namely thermolysis, Chelex-100, Chelex-100-triton X-100, using LAMP method. Only the Chelex-100 method could lyse 10^3^ CFU of *A. baumannii* competently, while all three methods successfully lysed 10^5^ CFU (Figure 5a). In comparison, both the Chelex-100 method and Chelex-100-triton X-100 method significantly outperformed thermolysis with 10^5^ CFU of *A. baumannii* (*p* ≤ 0.001), and no significant difference was observed between Chelex-100 and Chelex-100-tritonX-100 methods (Figure 5a). Based on the superior performance of the Chelex-100 method at the lower bacterial load (10^3^ CFU), it was selected for further optimization. To determine the optimal lysis time for *A. baumannii*, bacterial samples were treated with the Chelex-100 method for 5 min, 15 min, and 30 min. Compared to 5 min, extended processing time (15 min and 30 min) did not result in increased DNA release (Figure 5b). Therefore, a 5 min Chelex-100 method treatment was chosen for subsequent experiments.

### 3.6. Feasibility and Sensitivity Evaluation of the CLC Platform Using Simulated and Actual Air Samples

To assess the feasibility of the CLC platform for detecting CRAB in air samples, an airborne CRAB sample was collected using a wet cyclone sampler within an aerosol generation test chamber (Appendix A). The concentration of the simulated airborne CRAB was about 6 × 10^5^ CFU/mL, as determined by culture on nutrient agar. To determine the detection limit of the proposed CLC platform, the collected air sample was serially diluted tenfold to generate a series of bacterial suspensions from 6 × 10^1^ to 6 × 10^5^ CFU/mL. Subsequently, 1 mL of each diluted suspension was used to assess the sensitivity of the CLC platform. Both green and purplish-red fluorescence signals significantly increased at CRAB amounts ranging from 6 × 10^2^ CFU to 6 × 10^5^ CFU, but not from 6 × 10^1^ CFU and NC (Figure 6a,b). These results demonstrate that the LOD of the CLC platform for CRAB detection was as low as 6 × 10^2^ CFU per test.

To evaluate the feasibility of the CLC platform for detecting *A. baumannii* and the *bla*_OXA-23_ gene in actual air samples, five samples were collected from a tertiary hospital. Among these, only sample 1 was positive for *A. baumannii* and the *bla*_OXA-23_ gene, whereas the remaining four samples were negative (Figure 6c,d). tNGS confirmed the presence of *A. baumannii* in sample 1, with approximately 10^5^ copies/mL of *A. baumannii* genetic material detected. *A. baumannii* was not detected in samples 2 to 4 by tNGS. However, there were no data about the *bla*_OXA-23_ gene due to the absence of the *bla*_OXA-23_ target in the tNGS kit. To further validate the results, nucleic acids extracted from the five samples were amplified by PCR and sequenced using Sanger sequencing. The Sanger sequencing results showed that sample 1 was positive for the *bla*_OXA-23_ gene, while the other four samples were negative. Therefore, there was 100% consistency between the results of the CLC platform and the sequencing results (tNGS and Sanger sequencing), demonstrating the feasibility of the CLC platform for real air sample testing.

## 4. Discussion

LAMP has been widely applied in microbial nucleic acid detection due to its rapidity and operational simplicity [33,34,35]. However, conventional real-time fluorescence or colorimetric LAMP assays typically rely on non-specific dyes such as SYBR Green [36], hydroxy naphthol blue (HNB) [37], and Calcein [38]. Although these methods are straightforward and fast, the use of non-specific dyes limits their ability to discriminate between true-positive and false-positive signals, often resulting in relatively high false-positive rates [39,40]. Therefore, there remains a critical need to improve the specificity and overall detection accuracy of LAMP-based detection. Coupling LAMP with CRISPR-assisted detection offers a promising strategy to address this limitation and enhance the reliability of isothermal amplification assays. In this study, we developed a one-pot LAMP-CRISPR/Cas12a assay that combines the rapid amplification efficiency of LAMP with the sequence-specific recognition capability of the CRISPR/Cas12a system. The Cas12a-crRNA complex selectively binds to target double-stranded DNA sequences that are complementary to the crRNA, activating its trans-cleavage activity only in the presence of the specific target [41,42]. Consequently, the system generates strong fluorescence signals exclusively in true-positive reactions, while effectively eliminating false-positive readouts. In addition, the assay supports visual fluorescence readout under UV light, offering a simple, equipment-minimal approach suitable for on-site applications. Moreover, compared to the two-step LAMP-CRISPR-based detection method, the one-pot LAMP-CRISPR approach significantly reduces operational complexity and the risk of aerosol contamination from LAMP amplicons, thereby enhancing the overall accuracy and practicality of the detection platform.

Compared to other platforms and systems developed for the detection of *A. baumannii* and the *bla*_OXA-23_ gene, the CLC platform demonstrates favorable performance in terms of detection time, sensitivity, and equipment requirements. Although it may not be the best in all aspects, including detection time, limit of detection, and device portability, it achieves a good balance among these key parameters, making it capable of meeting the basic requirements for on-site detection (Appendix A).

Airborne pathogen detection is critically important due to its significant implications for public health. An ideal detection method or integrated platform should be simple, rapid, accurate, and user-friendly. The developed one-pot LAMP-CRISPR/Cas12a based platform, when coupled with a commercial wet cyclone air sampler, is capable of on-site detection of *A. baumannii* and the *bla*_OXA-23_ gene in air samples. For on-site visual detection, the minimum equipment required for the entire procedure, including DNA release and the one-pot LAMP-CRISPR/Cas12a assay, consists of pipettes, pipette tips, UV light, two heating blocks, and a handheld centrifuge. All the equipment could be integrated into a suitcase (Appendix A). Although the UV light, thermal block, and centrifuge require a line current source (~220 V), this power supply is readily available, making the CLC platform suitable for most settings. Additionally, the equipment used in the CLC platform has low power consumption (heating block 120 W; UV light and centrifuge 30 W each), making it compatible with commercial portable power stations. This further enhances the platform’s availability and convenience for on-site testing.

Our study focused on the detection of airborne *A. baumannii* and the *bla*_OXA-23_ gene. The simultaneous detection of both *A. baumannii* and the *bla*_OXA-23_ gene typically indicates the presence of carbapenem-resistant *A. baumannii* (CRAB). However, it is important to note that a positive result for *A. baumannii* and a negative result for the *bla*_OXA-23_ gene does not necessarily rule out the possibility of CRAB. This may occur if the strain belongs to a small number of carbapenemase subspecies, in which case further characterization using a multigene multiplex assay or other available diagnostic methods is required. The *bla*_OXA-23_ gene was chosen as the target for detection in this study due to its widespread distribution and common occurrence. Its global prevalence has been extensively documented [43,44,45]. Furthermore, a molecular epidemiological study conducted in Chinese hospitals demonstrated that 97.7% of CRAB isolates carried the *bla*_OXA-23-like_ gene, highlighting its high frequency in CRAB and its widespread dissemination across China [46].

Looking forward, we plan to broaden the scope of our platform to detect additional airborne pathogens, including common pathogenic bacteria (e.g., *Staphylococcus aureus, Legionella pneumophila*), viruses (e.g., influenza virus, coronaviruses), and fungi (e.g., *Cryptococcus neoformans*), and common antimicrobial resistance genes (e.g., *bla*_CTX-M_, *bla*_NDM_, *vanA, bla_OXA-58_*). These pathogens and resistance genes all have the potential to cause hospital-acquired infections. Furthermore, developing an integrated device that combines sample collection, DNA lysis and nucleic acid amplification, such as a microfluidic chip, would simplify processes and enhance convenience.

## 5. Conclusions

In this study, we developed a simple and rapid CLC platform capable of detecting *A. baumannii* and the most prevalent carbapenem resistance gene, *bla*_OXA-23_. By integrating a high-flow-rate wet cyclone sampler with high collection efficiency, the CLC platform enables seamless air sample detection, achieving a complete workflow from sample collection to result output. The platform was validated using both simulated and actual air samples collected from a hospital setting. The detection of simulated air samples showed the LOD of the established platform was about 6 × 10^2^ CFU of CRAB per test. Furthermore, for the detection of actual air samples, the results obtained through sequencing technology were fully consistent with those generated by the established CLC platform, thereby demonstrating the feasibility of the CLC platform. The entire detection process, from sample collection to results acquisition, required only about 70 min with minimum equipment. Moreover, the incorporation of fluorescent dye coding facilitated convenient and specific signal output for dual nucleic acid detection, making the platform highly suitable for on-site applications. This made our approach highly practical for on-site testing airborne *A. baumannii* and its carbapenem resistance gene *bla*_OXA-23_, particularly in environmental surveillance following a hospital outbreak of *A. baumannii*.

## Figures and Tables

**Figure 1 microorganisms-13-00976-f001:**
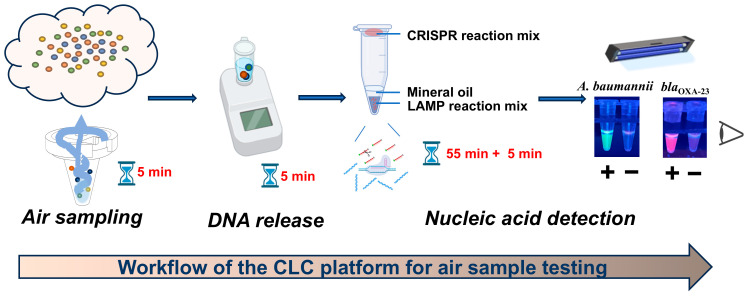
Workflow of the CLC platform for air sample testing.

**Figure 2 microorganisms-13-00976-f002:**
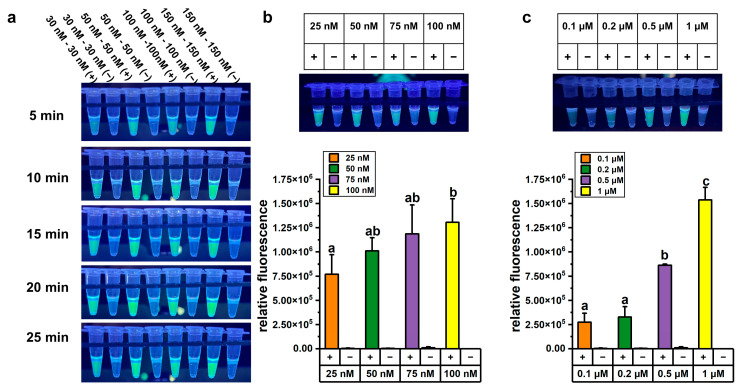
Optimization of the one-pot LAMP-CRISPR/Cas12a assay targeting *A. baumannii*. (**a**) Fluorescence changed over time in the one-pot LAMP-CRISPR/Cas12a assay with a 1:1 ratio of Cas12a and crRNA. (**b**) Fluorescence signals in the one-pot LAMP-CRISPR/Cas12a assays with varying concentrations of crRNA. (**c**) Fluorescence signals in the one-pot LAMP-CRISPR/Cas12a assays with varying concentrations of FAM-labeled reporter probe. All concentrations in this figure referred to final concentration in the total 50 μL LAMP-CRISPR/Cas12a assay, comprising a 25 μL LAMP reaction and a 25 μL CRISPR/Cas12a reaction system, excluding 20 μL of mineral oil. The error bars represent mean + standard deviation (*n* = 3 replicates). Bars labeled with different letters indicate statistically significant differences (*p* < 0.05). Shared letters (e.g., ‘a, ab’) denote no significant difference between groups.

**Figure 3 microorganisms-13-00976-f003:**
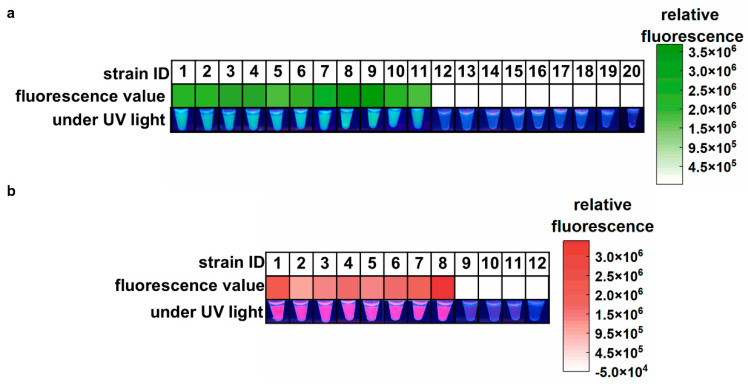
Feasibility and specificity of the one-pot LAMP-CRISPR/Cas12a assay. (**a**) Target: *A. baumannii*. 1, *A. baumannii* (ATCC 19606); 2, *A. baumannii* CI-1; 3, *A. baumannii* CI-2; 4, *A. baumannii* CI-3; 5, *A. baumannii* CI-4; 6, *A. baumannii* CI-5; 7, *A. baumannii* CI-6; 8, *A. baumannii* CI-7; 9, *A. baumannii* CI-8; 10, *A. baumannii* EI-1; 11, *A. baumannii* EI-2; 12, *Staphylococcus aureus*; 13, *Staphylococcus epidermidis*; 14, *Streptococcus pneumoniae*; 15, *Escherichia coli*; 16, *Pseudomonas aeruginosa*; 17, *Klebsiella pneumoniae*; 18, *Candida albicans*; 19, *Aspergillus fumigatus*; 20, *Cryptococcus neoformans*. (**b**) Target: *bla*_OXA-23_ gene. 1, *A. baumannii* CI-1; 2, *A. baumannii* CI-2; 3, *A. baumannii* CI-3; 4, *A. baumannii* CI-4; 5, *A. baumannii* CI-5; 6, *A. baumannii* CI-6; 7, *A. baumannii* CI-7; 8, *A. baumannii* CI-8; 9, *A. baumannii* (ATCC 19606); 10, CSAB 1; 11, CSAB 2; 12, CSAB 3. CI: clinical isolate; EI: environmental isolate; CSAB: Carbapenem-Susceptible *A. baumannii*; *A. baumannii* CI 1-8 were *A. baumannii* isolates that carried the *bla*_OXA-23_ gene. CSAB 1-3 were confirmed to be devoid of the *bla*_OXA-23_ gene.

**Figure 4 microorganisms-13-00976-f004:**
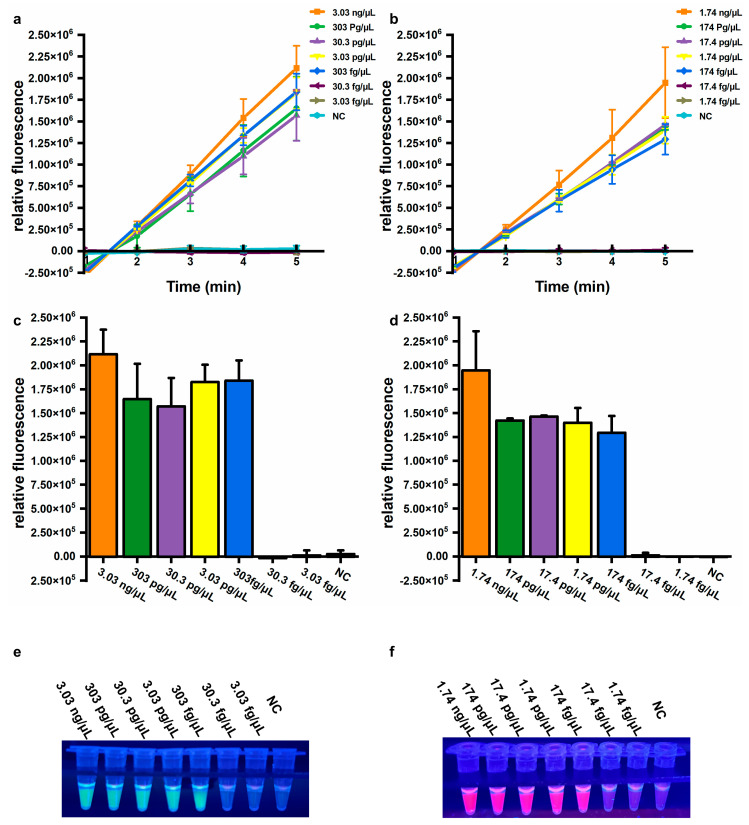
Sensitivity of the one-pot LAMP-CRISPR/Cas12a assay. (**a**,**b**) Relative fluorescence changed within 5 min. (**c**,**d**) Relative fluorescence values after 5 min of Cas12a-mediated cleavage. Error bars showed mean + standard deviation (*n* = 3 replicates). (**e**,**f**) Direct observation by the naked eye under UV light (312 nm). (**a**,**c**,**e**) Target: *A. baumannii*. (**b**,**d**,**f**) Target: *bla*_OXA-23_. NC, negative control.

**Figure 5 microorganisms-13-00976-f005:**
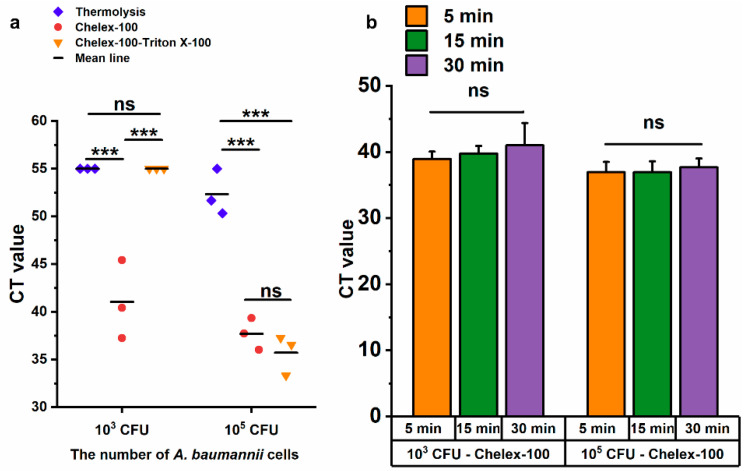
Evaluation of three DNA release methods by LAMP. (**a**) The cycle threshold (CT) values of the LAMP reaction using templates obtained by three methods after 30 min of treatment. (**b**) Comparison of treatment durations for 10^3^ CFU and 10^5^ CFU of *A. baumannii*. ns: no significance; *** *p* < = 0.001. Error bars represent mean + standard deviation (*n* = 3 replicates). The CT value of negative results was defined as 55.

**Figure 6 microorganisms-13-00976-f006:**
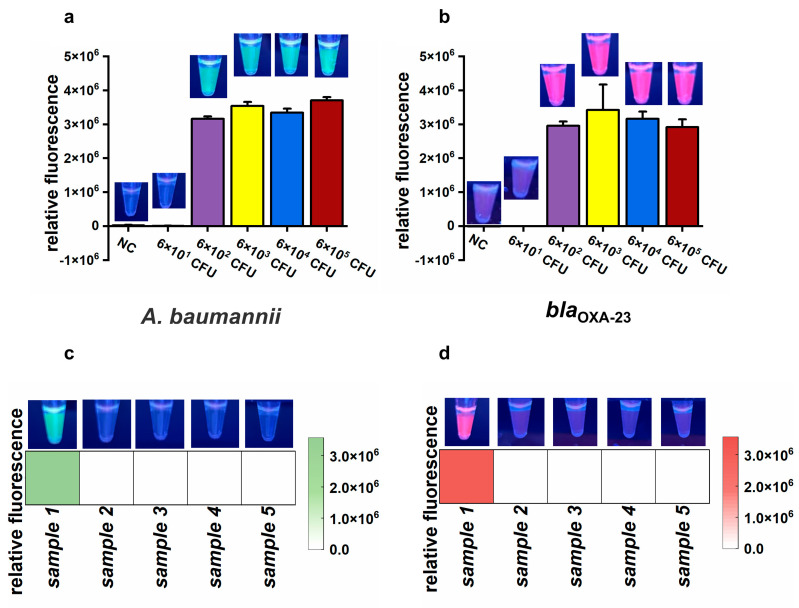
The results of the simulated and actual air samples. (**a**,**b**) Endpoint relative fluorescence and direct observation by the naked eye under UV light (312 nm) of simulated air samples. Error bars show mean + standard deviation (*n* = 3 replicates). (**c**,**d**) Final relative fluorescence and direct observation by the naked eye under UV light (312 nm) of five actual air samples. (**a**,**c**) Target: *A. baumannii*. (**b**,**d**) Target: *bla*_OXA-23_.

## Data Availability

The original contributions presented in this study are included in the Appendix A. Further inquiries can be directed to the corresponding authors.

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
