# Peer review of "On-Site Dual Detection of Airborne *Acinetobacter baumannii* and Its Carbapenem-Resistant Gene *bla*_OXA-23_ Using a One-Pot Visual LAMP-CRISPR/Cas12a-Based Platform"

_microorganisms, 2025, doi:10.3390/microorganisms13050976_

Round 1

Reviewer 1 Report

Comments and Suggestions for Authors

All in all I would think that the presented data show that the developed system are potentially interesting for those readers who work or are interested in clinical air quality control not only for A. baumannii but also for other airborne microorganisms that pose a great health risk for patients as well as staff in hospitals world wide. Authors should revise the text along the comments to the authors given below. I have also marked the comments and suggetions in the pdf-file that is attached to this evaluation.

Line 30: Key words should be chosen here which do not already appear in the title of the manuscript, at least not all of them.

Line 40: Citation Nr. 6 seems to be a miss-citation. This should probalbly read “World Health Organization WHO Bacterial priority pathogens list, 2024: bacterial pathogens of public health importance to guide research, development and strategies to prevent and control antimicrobial resistance. 2024 https://www.who.int/publications/i/item/9789240093461“

Line 49: Better use “infected“ instead of “colonized“?

Line 57: Please insert the words “Acinetobacter baumannii“ in citation No. 10 after “Survival of…“!

Line 73: Please insert the words “Actinobacillus pleuropneumoniae in citation No. 19 after “aerosolized ….“!

Lines 94-99: This part of the text should be deleted here since it provides information that is a result of the current study and must therefore be transfered to the results section, if not described there anyway.

Line 105: bacterial instead of bacteria, or delete the word “bacteria“ before the word “strain“, also in line 107.

Lines 109-110: Candida albicans, Cryptococcus neoformans and Aspergillus niger are not bacteria

Line 110: Please indicate whether or not the strains from the auhors´ lab collection are publically available and how these were identified.

Line 124: Using a citation here indicates that the primers used in the current study were taken from the cited literature. However, since the used primers are different from the cited paper, I suggest to delete the citation at this point.

Line 131: “… the details are presented“ instead of “… the details were presented“

Line 135: delete “a“ before “20 µL“

Line 144: Better to phrase this like “… reaction mixture was placed on the inner surface of the lid oft he reaction tubes.“

Line 182: Delete “LAMP reaction to the“ before “LAMP …“

Line 193: Authors state that the bioaerosol test chamber was thoroughly sterilized. Please describe the sterilization process in more detail, e.g. which disinfectant was used in which amount and how long was the incubation time.

Line 207: Please exchange the part of the sentence describing the test kit with more detailed information: Respiratory Pathogen Detection Kit (KS608-100HXD96, KingCreate, Guangzhou, China)

Line 208: Please provide more information abtout the analysis, e.g. what was sequenced and which platform was used for the analysis.

Line 210: replace the word “were“ with the word “are“

Line 214: replace the word “was“ with the word “is“

Line 293: Authors state, with reference to figure 3, that 10 common clinical isolates were teste das non-target organisms. However, Fig. 3a shows only 9 non-Acinetobacter isolates. Please check!

Line 311: delete the word “that“

Line 350: Figure 5, Please describe the symbols used in figure 5a!

Line 359: “concentration“ instead of “concentrations“

Line 361:“was“ instead of “were“

Line 373: What was the sensitivity of the tNGS method? Would copy numbers that are below the LOD of the LAMP assay have been detected here? Please discuss!

Line 376: Please give information about the primers used to prepare the sequencing target and put them into table S1 if not already there. Also, the resulting sequence should be made publically available in GenBank and the accession number put in the manuscript.

Line 384: “show“ instead of “showed“

Lines 397-398: I wonder whether the equipment used here was all battery operated or if rather a line current source was necessary for their operation. Please give those details in the materials and methods part.

Lines 387-409: General remark: I miss a critical discussion of the consequences that a positive result of the developed test in a clinical setting might have or should have. Especially in cases in which the test for CRAB is positive but the test for the blaOXA-23 is neagative. Should there be further analyses to test for the presence of other mechanisms of resistance which, according to the authors, exist? Or are those mechanisms infrequent enough as to declare a blaOXA-23 negative A. baumannii strain to be of no relevance for patients or for the clinical environment?

Reviewer 2 Report

Comments and Suggestions for Authors
  1. Hu et al. introduced a CLC platform that includes DNA release, amplification, and on-site detection. The manuscript data is well-organized and high quality. However, some minor points need to be thoroughly revised before publication.

    1. Introduction: There is no introduction or reason why Chelex was chosen for DNA capture and release. Also, the mechanism of DNA release should be mentioned where applicable.
    2. Also, there was no introduction to LAMP or isothermal amplification for airborne. These points should be added to the introduction to clarify the purpose of this research. 
    3. Although the research concentrated on on-site detection using the Cas system. However, it also needs to show some gel image results for double-checking and confirmation. 
    4. What would happen if Chelex was not used for DNA released after heat treatment (63 deg)? Also, is 5 minutes enough for heat lysis? The authors suggest checking this approach with more complex bacteria samples, which could affect performance. Also, Chelex may inhibit the LAMP reaction at high concentrations. So, this data should also be included in the revision to prove the method.
    5. There is a major concern that the authors suggested in building an approach for on-side detection DNA, which kind of heater was used for controlling the LAMP, heat lysis, and how big the UV light is for detection. This kind of concern should be fully addressed clearly in conclusions if it could hinder the need for this approach.
    6. How many times have the experiments been repeated? The authors suggested adding the information in each image.
    7. Fig. 3 is poor in the image, which requires identical image resolution and size to make better visualizing images.
    8. In Fig. 4, the graph lines were hard to see. It is better to change to the simple color system, which improves the visualization. It is the same comments for Fig. 5

Reviewer 3 Report

Comments and Suggestions for Authors

This manuscript describes investigations of A. baumannii from air samples. Topic of this manuscript is an interesting issue however, some parts in the text are unclear.

1) In the conclusion part a short comparative analysis should be given, to demonstrate whether the detection limit of this technique (6x102CFU/) is better than that of other available technique.

2) In the materials and methods part it is mentioned, that "Staphylococcus aureus, Staphylococcus epidermidis, Streptococcus pneumoniae, Escherichia coli, Pseudomonas aeruginosa, Klebsiella pneumoniae, Candida albicans, Aspergillus fumigatus, and Cryptococcus neoformans strains were obtained from our lab collection"

However, were these clinical isolates, or control strains ? What is the origin of these strains?

3) Materials and methods, please, revise this:  "The bacteria strain A. baumannii (ATCC 19606) was..." proper form: A. baumannii (ATCC 19606) control strain was...

4) Figure 4 should be enlarged. It is difficult to see the values on it.

Round 2

Reviewer 1 Report

Comments and Suggestions for Authors

The text of the manuscript has very much improved by the changes made. All my comments and suggetions for text editing have been followed. I have made only a few more corrections that can be found in the manuscript file added to this review report (see lines 30, 83, 445). My judgement is that the text is ready for publication after these very minor changes have been made.

Author Response

I have carefully reviewed the feedback but could not locate the specific revision suggestions from Reviewer 1 in the materials provided.

Reviewer 3 Report

Comments and Suggestions for Authors

This manuscript has been revised. All necessary modifcations have been done in the text.

Comments on the Quality of English Language

The quality of English is good.

Author Response

We sincerely appreciate your time and valuable comments, which have significantly improved our manuscript.